# Resveratrol and Its Analogue 4,4′-Dihydroxy-trans-stilbene Inhibit Lewis Lung Carcinoma Growth In Vivo through Apoptosis, Autophagy and Modulation of the Tumour Microenvironment in a Murine Model

**DOI:** 10.3390/biomedicines10081784

**Published:** 2022-07-25

**Authors:** Monica Savio, Alessandra Ferraresi, Chiara Corpina, Sara Vandenberghe, Chiara Scarlata, Virginie Sottile, Luca Morini, Beatrice Garavaglia, Ciro Isidoro, Lucia Anna Stivala

**Affiliations:** 1Immunology and General Pathology Unit, Department of Molecular Medicine, University of Pavia, 27100 Pavia, Italy; monica.savio@unipv.it (M.S.); chiara.corpina01@universitadipavia.it (C.C.); sara.vandenberghe01@universitadipavia.it (S.V.); chiara.scarlata01@universitadipavia.it (C.S.); virginie.sottile@unipv.it (V.S.); 2Laboratory of Molecular Pathology, Department of Health Sciences, University of Piemonte Orientale “A. Avogadro”, 28100 Novara, Italy; alessandra.ferraresi@med.uniupo.it (A.F.); beatrice.garavaglia@uniupo.it (B.G.); 3Unit of Legal Medicine, Department of Public Health, Experimental and Forensic Medicine, University of Pavia, 27100 Pavia, Italy; luca.morini@unipv.it

**Keywords:** nutraceuticals, stilbenes, tumour microenvironment, immune cells, cancer-associated fibroblasts, autophagy, cell proliferation, angiogenesis

## Abstract

Lung cancer is the most prevalent cancer worldwide. Despite advances in surgery and immune-chemotherapy, the therapeutic outcome remains poor. In recent years, the anticancer properties of natural compounds, along with their low toxic side effects, have attracted the interest of researchers. Resveratrol (RSV) and many of its derivatives received particular attention for their beneficial bioactivity. Here we studied the activity of RSV and of its analogue 4,4′-dihydroxystilbene (DHS) in C57BL/6J mice bearing cancers resulting from Lung Lewis Carcinoma (LLC) cell implantation, considering tumour mass weight, angiogenesis, cell proliferation and death, autophagy, as well as characterization of their immune microenvironment, including infiltrating cancer-associated fibroblasts (CAFs). C57BL/6J mice started treatment with RSV or DHS, solubilised in drinking water, one week before LLC implantation, and continued for 21 days, at the end of which they were sacrificed, and the tumour masses collected. Histology was performed according to standard procedures; angiogenesis, cell proliferation and death, autophagy, infiltrating-immune cells, macrophages and fibroblasts were assessed by immunodetection assays. Both stilbenic compounds were able to contrast the tumour growth by increasing apoptosis and autophagy in LLC tumour masses. Additionally, they contrasted the tumour-permissive microenvironment by limiting the infiltration of tumour-associated immune-cells and, more importantly, by counteracting CAF maturation. Therefore, both stilbenes could be employed to synergise with conventional oncotherapies to limit the contribution of stromal cells in tumour growth.

## 1. Introduction

Lung cancer is responsible for most cancer deaths in both men and women throughout the world. It is among the top three cancer types for incidence and the first in terms of mortality [1]. According to data collected by the Agency for Research on Cancer (available via the Global Cancer Observatory platform) [2], lung cancer includes very different types and subtypes, which affects the strategies for prevention, early detection, diagnosis, and clinical management. Despite advances in treatment modalities, the therapeutic effects are still poor, and the 5-year overall survival rate of lung cancer of all stages combined is no more than 18% (57.4% in early stage versus 5.2% with distant metastasis at diagnosis) [3]. Based on the currently available therapeutic technologies, most patients cannot be cured for the emergence of chemoresistance, leading to a poor outcome in the majority of metastatic patients [4]. This calls for further research efforts to explore new personalised treatments, develop novel combination of therapies exploiting any synergies, and explore innovative strategies targeting the tumour microenvironment. In fact, in addition to tumour-intrinsic factors, the tumour microenvironment exerts a key influence in every step of tumorigenesis [5,6]. The continuous and dynamic cross-talk between the numerous cellular components within a tumour mass is highly complex and crucial in shaping tumour architecture. For this reason, the analysis of the microenvironment factors, in addition to cancer cells, has become fundamental to predict response to treatment. Accordingly, interference with any component of the tumour-surrounding stroma may offer the opportunity to counteract its growth.

A wide range of in vivo studies have proved that resveratrol (3,5,4′-trihydroxy-*trans*-stilbene, RSV) may exert anti-cancer properties (reviewed in [7,8,9]) via multiple mechanisms and pathways [10,11,12,13,14,15,16]. Additionally, it has been reported that RSV’s activity could also be associated with specific microenvironmental changes, playing important anti-inflammatory [17,18] and immunosuppressive roles [19,20,21], blocking the cross-talk between cancer cells and stromal cells [22], inhibiting tumour angiogenesis [23,24], and cancer-associated fibroblast (CAFs) activation [25,26].

In comparison to RSV, very few in vivo studies have investigated those RSV derivatives which have shown a stronger anti-cancer activity in in vitro models, such as methoxylated [27,28], hydroxylated [29], halogenated derivatives [30,31]. Among the different hydroxylated RSV analogues, we have been studying 4,4′-trihydroxy-*trans*-stilbene (DHS) for many years because of its anti-oxidant [32,33,34], cytotoxic and apoptosis-inducing properties higher than that of the parental molecule [35], as well as its effects on cancer cell proliferation [36,37] and angiogenesis [36,38]. In our previous research, we used a Lewis Lung Carcinoma (LLC) mouse model to study the tumour response to DHS. The size of the primary LLC tumour and metastasis were strongly inhibited by the treatment causing a suppression of cell proliferation and angiogenesis [37]. Moreover, a recent study has demonstrated that DHS prevented colon tumour growth by counteracting tumour-associated macrophages’ (TAMs) M2 differentiation and PD-1 activation in the tumour microenvironment [39].

In this work, we present further insight into the pleiotropic mechanisms underlying the anti-tumour properties of DHS and its parental molecule RSV, in C57BL/6J mice bearing cancers resulting from LLC implantation. We present a comprehensive analysis of tumours formed in mice treated with the two molecules, in terms of tumour mass weight, angiogenesis, cell proliferation and death, autophagy, as well as the characterization of their immune microenvironment, including the infiltrating lymphocytes (TILs), TAMs, and CAFs.

Our study provides the evidence that both the stilbenes are able to contrast the growth of the LLC tumour masses by inhibiting the tumour-supporting features promoted by the components of the tumour microenvironment. In detail, the phenotypic effects achieved by DHS treatment are more pronounced compared to that observed in RSV-treated mice. Taken together, our data may provide the rationale for the use of RSV analogues to synergise with conventional treatment to limit the participation of the tumour microenvironment in the growth of tumour mass and improve the outcome of lung cancer patients.

## 2. Materials and Methods

### 2.1. Reagents and Cell Culture

4,4′-dihydroxy-*trans*-stilbene (DHS) and resveratrol (RSV) were purchased from Santa Cruz Biotechnology. Murine Lewis Lung Carcinoma (LLC) cells, provided by the Experimental Zooprophylactic Institute of Brescia, Italy, were cultured in D-MEM supplemented with 8% FBS, 200 mM L-glutamine, 100 IU/mL penicillin, and 100 μg/mL streptomycin, all obtained from Thermo Fisher Scientific, Waltham, MA, USA.

### 2.2. Antibodies and Reagents

The primary antibodies used for immunohistochemistry (IHC) were the following: anti-PCNA mouse monoclonal antibody (PC10, Dako, Agilent Technologies, Santa Clara, CA, USA, 1:200), anti-CD31 rat monoclonal antibody (e-Bioscience, Thermo Fisher Scientific, Waltham, MA, USA, 1:400), anti-CD163 RabMab rabbit mAb (Abcam, Cambridge, UK, Clone:EPR19518, 1:500), anti-CD3 rabbit polyclonal antibody (Origene, Rockville, MD, USA, TA354250, 1:200) and anti-CD68 rat monoclonal antibody (Origene, Rockville, MD, USA, SM1550PS, 1:200) were used. Anti-LC3 rabbit polyclonal antibody (Sigma Aldrich, St. Louis, MO, USA, L7543, 1:1000), anti-p62 mouse monoclonal antibody (Millipore, Merck Darmstadt, Germany, MABC32, 1:500), anti-LAMP1 mouse monoclonal antibody (BD, Franklin Lakes, NJ, USA, 555798, 1:1000), pan-cytokeratin (pan-CK) mouse polyclonal antibody (Invitrogen, Thermo Fisher Scientific, Waltham, MA, USA, MA5-13203, 1:100), and α-SMA mouse monoclonal antibody (Sigma Aldrich, St. Louis, MO, USA, A5228, 1:250) were used for autophagy assessment in cancer cells and cancer-associated fibroblasts. Alexa Fluor 488 goat anti-rat secondary antibody was used for whole mount staining of CD31. Peroxidase-conjugated IHC secondary antibodies were purchased from Bethyl Laboratories, Montgomery, TX, USA. Alexa Fluor^TM^ 555 goat anti-mouse IgG (red fluorescence; A32727, 1:1000) or Alexa Fluor^TM^ 488 goat anti-rabbit IgG (green fluorescence; A32731, 1:1000) from Life Technologies, Thermo Fisher Scientific, Waltham, MA, USA, were used for immunofluorescence. The FragEL DNA fragmentation Detection kit TdT enzyme (Calbiochem, San Diego, CA, USA, QIA33) was applied to evaluate apoptosis.

### 2.3. Murine Tumour Model and Treatments

The experimental protocol used to treat animals is similar to the one we have previously published [37]. Briefly, 20 male C57BL/6J mice (4-weeks-old) purchased from Harlan Laboratories (Udine, Italy) were housed (Centro Interdipartimentale di Servizio per la Gestione Unificata delle Attività di Stabulazione e di Radiobiologia) and maintained under standard conditions of a 12 h dark/12 h light cycle, at a temperature of 24 ± 2 °C, and relative humidity of 50 ± 10%. DHS (25 mg/kg/day) or RSV (125 mg/kg/day) was added to drinking water (1% Et-OH/water) and replaced three times a week. All experimental procedures were performed in accordance with the European Convention for Care and Use of Laboratory Animals, and were approved by the local Animal Ethics Committee of the University of Pavia (Document n. 1, 2012). Mice were divided into 3 groups: vehicle (mice taking 1% EtOH, 6 animals), mice drinking DHS (9 animals), and mice drinking RSV (5 animals). Treatments started a week before injecting tumour cells, and continued until sacrifice. A single-cell suspension (1 × 10^6^ cells in 400 µL of saline buffer) was implanted subcutaneously in the left side of each animal. Three weeks later, the animals were killed by a lethal dose of ether, then tumour masses were collected, dissected, and fixed with 4% formaldehyde in phosphate buffered saline (PFA) (Carlo Erba, Rodano, MI, Italy) for histological analysis. Primary tumours were measured with a calliper, and their volume calculated according to a standard formula (length × width^2^ × 0.52) [40]. The determination of analytes in plasma was performed through a LC-MS/MS method, based on the same procedure previously published by Muzzio and co-authors [41]. Briefly, 1 mL acetonitrile, containing the internal standard (mefenamic acid) was added to 100 µL plasma, then was vortexed and centrifuged. The supernatant was removed, dried under nitrogen stream, and reconstituted in a water/methanol (4:1) mixed solution. The sample was injected in an LC-MS/MS system (Agilent 1100:1200 series HPLC, Agilent Technologies, Palo Alto, CA, USA, coupled with an AB Sciex 4000QTrap triple Quad, AB SCIEX, Foster City, CA, USA), operating in Multiple Reaction Monitoring (MRM) mode in negative polarization. The chromatographic separation was performed in reversed phase, gradient elution, through a C18 column (kinetex C18 column 100 × 2.1 mm i.d., 5 µm particle size, Phenomenex, Castelmaggiore, BO, Italy). Two different MRM transitions were selected.

### 2.4. Histology and Immunohistochemistry

Tissue samples were stained with H&E using a standard protocol (haematoxylin and eosin G, Sigma Aldrich and Merck Certistain, respectively). PCNA staining, as proliferation marker, was performed using M.O.M.^TM^ reagent kit (Vector Laboratories, Newark, CA, USA). CD163, CD68, and CD3 were detected on paraffin-embedded sections using standard immunohistochemistry protocols. In brief, tissue sections were deparaffinised in xylene and rehydrated in a series of concentrations of ethanol. Thereafter, the sections were treated to block endogenous peroxidase activity with 3% H_2_O_2_ for 45 min. Then, antigen retrieval was performed by microwaving sections in unmasking solution (Vector Laboratories, Newark, CA, USA) for 20 min. Non-specific binding was blocked by 10% goat serum in 1× phosphate-buffered saline (1× PBS) for 30 min. Sections were incubated with the primary antibodies at room temperature for 1 h in a moisture chamber. After that, the sections were incubated with the respective peroxidase-conjugated secondary antibodies. The colour was developed with diaminobenzidine (DAB SK-4100 kit, Vector Laboratories). The sections were counterstained with haematoxylin for 15 min, embedded with Eukitt (O. Kindler GmbH, Freiburg, Germany), observed by Nikon digital microscope Eclipse E80i, and image analysis was performed using Fiji-ImageJ software.

For autophagy assessment, samples were co-stained for LC3-p62, LC3-LAMP1, LC3-panCK, and LC3-α-SMA. Tissue sections were deparaffinised as described above, then antigen retrieval was performed by microwaving sections in 0.05 M EDTA solution (pH 8.0) for 30 min. Tissue sections were pre-blocked with 3% BSA + 0.05% NH_4_Cl in 1× PBS for 15 min and then blocked in 3% BSA in 1× PBS for 1 h. Sections were incubated with the primary antibodies overnight at 4 °C in a moisture chamber. After that, the sections were washed three times in 0.1% Triton-PBS and then incubated with the respective fluorochrome-conjugated secondary antibodies for 1 h at room temperature in a moisture chamber. Nuclei were stained with the UV fluorescent dye DAPI (4′,6-diamidino-2-phenylindole). Sections were washed again as described previously and then mounted using SlowFade reagent (Life Technologies, Thermo Fisher Scientific, Waltham, MA, USA, S36936). Samples were imaged under a fluorescence microscope (Leica DMI6000, Leica Microsystems, Wetzlar, Germany). Quantification of fluorescence intensity was performed with the software ImageJ, following the instructions for quantification of single-channel or double-channel fluorescence intensity from https://theolb.readthedocs.io/en/latest/imaging/measuring-cell-fluorescence-using-imagej.html, accessed on 15 January 2022.

### 2.5. Whole Mount Staining of CD31

For whole mount staining of CD31, fresh tumour tissues were harvested, fixed in 4% PFA and processed as previously described [37]. Confocal microscopy (Leica TCS SP5 II) images of 6–10 randomised fields were collected and analysed using Adobe Photoshop CS4 software, 11.0 adobe California.

### 2.6. DNA Fragmentation Assay

Terminal deoxynucleotidyl transferase dUTP nick end–labelling (TUNEL) colorimetric assay (Calbiochem, San Diego, CA, USA, QIA33) was performed in mouse tumour masses, according to the instructions, to measure DNA fragments in vivo. The apoptotic index (AI) was calculated as number of apoptotic cells/masses.

### 2.7. Statistical Analysis

Statistical analysis was performed using GraphPad Prism 5.0 software. Bonferroni’s multiple comparison test after one-way ANOVA analysis (unpaired, two-tailed) was employed. Significance was considered as follows: **** *p* < 0.0001; *** *p* < 0.001; ** *p* < 0.01; * *p* < 0.05. Experimental data were presented as mean determinants ± S.D.

## 3. Results

### 3.1. LLC-Tumour Growth and Size Are Decreased in DHS- and RSV-Treated Mice

We investigated and compared the anticancer activity of DHS and RSV in LLC bearing-mice treated with either compound, compared to vehicle controls, for 21 days. During this time, neither acute toxicity nor side effects, such as lethargy and sickness, were detected in treated mice. No significant changes were observed on either mouse survival or body weight in the animals of the three experimental groups. Local tumour growth was monitored every day for 3 weeks, and primary masses were explanted and measured with a calliper at the end of each treatment. As shown in Figure 1a,d, the mean LLC-tumour volumes in DHS- and RSV-treated groups were significantly decreased by about 63% and 52% compared to the vehicle group (*p* < 0.05), respectively. A similar reduction in tumour weight of treated mice was observed, although not statistically significant. Interestingly, despite morphological similarity between tumour tissue specimens from vehicle- and stilbenes-treated mice, a much more prominent necrosis was found in the vehicle group (Figure 1b, n) than that observed after the treatment with DHS or RSV. Next, we carried out the immunostaining for PCNA, an endogenous cell proliferation marker [42], to assess the growth inhibitory ability of DHS and RSV in LLC-tumours in vivo (Figure 1c,d)). Compared to the vehicle group masses, both DHS and RSV reduced the number of PCNA-stained positive cells, though significantly only in the DHS-treated group with a 30% reduction (*p* < 0.05).

### 3.2. Plasma HPLC-MS/MS Detection of RSV

RSV was detectable in mice plasma at the end of the treatment achieving a concentration of 11.8 ng/mL as shown in Figure 2. Though the MRM chromatograms of real plasma samples showed some interferences, potentially due to degradation products of RSV or metabolites, the peak of RSV provided an adequate shape and satisfied all the criteria for identification and quantification.

### 3.3. Angiogenesis Is Reduced in LLC-Tumours of DHS- and RSV-Treated Mice

Angiogenesis is an important hallmark of cancer aggressiveness, since it not only promotes cancer cell growth and survival but also supports metastasis formation [43]. To investigate the in vivo anti-angiogenic effects of DHS and RSV treatments, we performed whole mount immunofluorescence staining of CD31, a marker highly expressed in endothelial cells exhibiting an angiogenic phenotype. Three-dimensional confocal images in Figure 3a showed a higher blood vessel density and integrity in the vehicle-treated group than in those treated with DHS or RSV. Analysis of vascular density detected by CD31 staining highlighted a significant reduction of about 70% in the DHS-treated group in comparison with the vehicle group (Figure 3b); a reduction of about 36%, although not statistically significant, was also observed in RSV-treated group. These results demonstrated that DHS inhibits tumour angiogenesis in vivo more efficiently than RSV.

### 3.4. The Treatment with DHS or RSV Decreases the Infiltrating Immune Cells in LLC-Tumour Microenvironment

Tumours are characterised by a complex cross-talk between malignant cells and many normal cell types that can influence cancer behaviour during its progression [44]. In particular, infiltrating immune cells can either induce anti-tumour immune responses or promote tumour growth and progression depending on their functional subtype [45]. To better understand whether and to which extent DHS and RSV may affect immune cell populations in LLC-mass microenvironment, we evaluated the presence of tumour-associated macrophages (TAMs) and infiltrating lymphocytes (TILs) by IHC techniques. CD163 (M2-like TAMs), CD68 (TAMs in general), and CD3 (total T cells) were used as markers to distinguish the different subtypes. Representative images of CD163-labelled cells in vehicle, DHS, and RSV specimens are shown in Figure 4a, and at higher magnification in vehicle sample in Figure 4b.

Tumour cell positivity was also observed for CD68 and CD3 (Appendix A), and the subsequent quantification analysis (Figure 4c) clearly indicated that both stilbenes can induce a statistically significant decrease of all the immune cell types considered, as compared to vehicle tumour masses. DHS reduced the presence of both M2-like TAMs (*p* < 0.05), as well as of total TAMs and TILs (*p* < 0.01) inside the tumour masses. Similarly, RSV induced a decrease in the M2-like macrophages number (*p* < 0.05), total macrophages, and T lymphocytes (*p* < 0.001).

### 3.5. LLC-Tumours in DHS- and RSV-Treated Mice Display a Reduction of Necrosis in Parallel with Increased Apoptosis

Cell death, either in terms of necrosis and apoptosis, can affect the growth and volume of tumours after treatment with antitumoral compounds. Since we observed a different size of LLC tumour masses depending on the treatment (Figure 1a), we decided to determine in histological sections whether and to which extent necrosis and apoptosis were present. Necrosis appeared quantitatively greater in vehicle tumour masses than in those treated with DHS or RSV, with a reduction in the latter of 92% (Figure 5a). Small vessels in formation were well detectable in RSV masses, less in those of the DHS-treated group. Furthermore, using a FragEL DNA fragmentation detection kit, we evaluated apoptotic cells inside the tumour masses (Figure 5b). Many apoptotic events were detected in the necrotic tumour area. We quantified apoptosis within the tumour masses. Apoptotic cells were significantly increased in LLC masses after treatment with both stilbenes. In DHS and RSV groups, 55% of apoptotic cells was estimated compared to the 25% estimated in the control group (representative panels and quantitative analysis shown in Figure 5b).

### 3.6. DHS- and RSV-Treated Tumours Exhibit Autophagy Upregulation and Reduced CAFs Population

Autophagy is a catabolic process that governs cell homeostasis by ensuring macromolecular turnover, and it plays a role in reshaping of the tumour microenvironment [46]. We assessed the autophagy flux by monitoring LC3 (autophagosome marker) and p62 (whose turnover is representative of autophagic clearance) levels, and the formation of autolysosomes indicated by the co-localization of LC3 and LAMP1 (lysosomal marker) (Figure 6).

We have previously reported that resveratrol could interrupt the cross-talk between cancer-associated fibroblasts (CAFs) and cancer cells by autophagy modulation [25]. Here we monitored autophagy in the two cell populations distinguished by pan-cytokeratin (epithelial marker) and α-SMA (marker of activated fibroblasts) staining. As shown in Figure 7a, stilbenes-treated tumour masses displayed an upregulation of LC3 levels in cancer cells compared to that observed in the vehicle group. Notably, both DHS and RSV greatly abrogated the activation of fibroblasts, as indicated by the significant downregulation of α-SMA (*p* < 0.0001 and *p* < 0.001, respectively). Of note, these phenotypic changes were associated with the increased formation of LC3-positive spots, suggesting that autophagy may prevent the phenoconversion of normal fibroblasts to CAFs (Figure 7b). Importantly, these effects were more pronounced in DHS- compared to RSV-treated masses.

## 4. Discussion

Extensive research over many years has revealed the beneficial potential of natural plant-derived compounds, and among these, the stilbenoids found in several plant species [47], known to exert multiple biological activities (reviewed in [48]). We have previously demonstrated that DHS can inhibit proliferation as well as invasion and metastasis of murine Lewis lung carcinoma in vivo [37], in full agreement with other in vivo reports, albeit in studies performed with different types of cancer cells [38,39,49,50]. Additionally, we have shown that RSV can exert anti-neoplastic activity via non-coding RNAs’ epigenetic modulation of the pathways governing cell homeostasis, cell proliferation, cell death, and cell motility [51]. In the present work, we have compared the anti-tumour properties of DHS with those of its parental molecule RSV, the most actively investigated stilbene, in the same LLC syngeneic mouse model [37]. Our data clearly show that both stilbenes have anticancer properties, reducing significantly (*p* < 0.05) the volumes of LLC masses with respect to the vehicle-treated mice. In search for the mechanisms underlying such effect, DHS confirmed its antiproliferative and antiangiogenic effect [37,38]. RSV elicited similar effects, though in our experimental model, it did not reach statistical significance as instead reported by Kimura and collaborators [24,38]. Besides differences in the cancer cells inoculated in mice (colon 26 vs. Lewis lung), such discrepancy is likely due to differences in the route of stilbene administration (intraperitoneally vs. daily drinking) and treatment duration, that are known to affect RSV concentration in tissues and organs. In this study, the RSV plasma levels, as determined by HPLC/UV/MS at the end of the administration period by drinking water, were 12 ng/mL, that is higher than 5 ng/mL detected in DHS-treated mice [37]. It is highly probable that this difference depends in part on the doses used for the animal treatments, i.e., 25 mg/kg/day for DHS vs. 125 mg/kg/day for RSV, which was chosen to achieve a higher plasma concentration compared to that previously found for DHS. Nevertheless, both plasma concentrations were lower than expected. It is worth noting that each animal took in small water amounts (2–2.5 mL/day), and consequently of each compound. In addition, RSV is extensively metabolised to resveratrol-3-sulfate and resveratrol-3-O-β-D-glucoronide and rapidly excreted [52,53]. The better pharmacokinetic profile of DHS compared to that of RSV [54] may also contribute to the lack of correlation between the administered doses and the respective plasma levels found.

Despite the higher plasma concentration of RSV, DHS exerted a stronger anticancer activity than its parental molecule, in accordance with our study [37] and other previous studies [33,38]. The anticancer properties of many bioactive compounds include apoptosis, necrosis, and autophagy. Both compounds studied in this work significantly induced apoptotic events in the tumour LLC masses, whereas a reduction in necrosis was observed in comparison with vehicle-treated masses. This may explain why macroscopically, concomitantly with a reduced angiogenesis, the tumour masses of RSV- or DHS-treated mice appeared less soft and bloody to the touch than those of the vehicle-treated sample. The ability of RSV to induce apoptosis in in vivo models has been reported (reviewed in [8]). However, there are no data on the ability of DHS to reduce necrosis in LLC masses, as we show in the present work. Considering these findings, we investigated whether the stromal and immune cells surrounding and infiltrating the LLC masses would influence their macroscopic feature and, at the same time, play a key role in the anticancer activity of DHS and RSV. A significant reduction in tumour-associated macrophages (CD68+), including the “pro-tumour” M2 phenotype (CD163+), together with lymphocytes (CD3+) was detected in LLC masses after treatment with both the stilbenes. In full agreement with previous data [55,56], these findings indicate that both compounds exert pleiotropic effects to prevent or reduce cancer growth by affecting several tumour-promoting hallmarks, i.e., cancer cell proliferation, angiogenesis, and M2 macrophages. Similarly, both stilbenes are able to markedly abrogate fibroblast activation to CAFs, thereby interrupting their crucial role in the cross-talk supporting the reprogramming towards a permissive tumour microenvironment. We have previously demonstrated the ability of RSV to interrupt pro-tumour communications between CAFs and cholangiocarcinoma cells by induction of autophagy [25]. Beyond self-eating and recycling damaged organelles, autophagy can influence dynamic cellular processes and lead to tumour microenvironment reprogramming [57]. Here, we found a statistically significant increase of autophagy markers by RSV, and to a greater extent by DHS, in LLC primary masses. The induction of autophagy by RSV has been reported in many in vitro studies, and here we have confirmed such effect in a syngeneic mouse model of lung cancer. Remarkably, we have previously demonstrated that RSV-induced autophagy can oppose ovarian cancer cell migration and proliferation while supporting a dormant phenotype [58,59,60].

This is the first report revealing that DHS induces both apoptosis and autophagy. Upregulation of autophagy has been reported to impair EMT and to halt cancer cell migration [61]; this observation is in support of the reduction in liver metastatic lesions by DHS treatment in both mice and zebrafish tumour models [37], and in a colon cancer model [56]. Altogether, these findings underline the anticancer potential of DHS and its higher bioactivity compared to RSV. Considering the growing interest in targeting the tumour microenvironment as a promising strategy for cancer treatment, RSV and even more, DHS, present promising anticancer activity that should be further investigated. Targeting the tumour microenvironment is a very attractive therapeutic approach for the treatment of cancer, and therefore RSV and DHS may represent an effective adjuvant strategy to be used to complement conventional oncotherapies.

## 5. Conclusions

Taken together, our findings reveal that the administration of RSV and, in more pronounced way, of its analogue DHS, reduces the growth of the LLC tumour masses in C57BL/6J mice. This antitumour effect can be attributed to an increase of apoptosis and autophagy, and an inhibition of the tumour-supporting features promoted by the components of the tumour microenvironment. This preclinical evaluation provides the rationale for future studies to explore whether RSV analogues may synergise with conventional treatments to limit the tumour microenvironment’s support to cancer growth, and improve the therapeutic outcome for lung cancer patients.

## Figures and Tables

**Figure 1 biomedicines-10-01784-f001:**
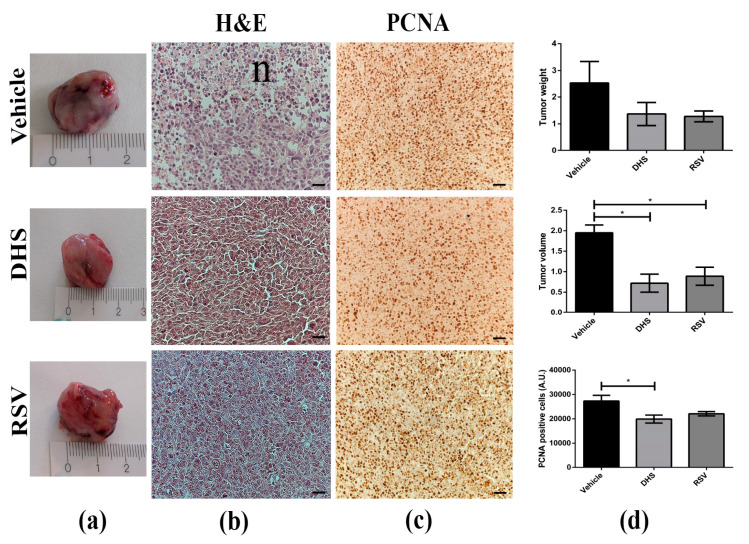
(**a**) Representative macroscopic view of LLC primary tumours in control (vehicle, 6 mice) and 4 weeks DHS- and RSV-treated mice (9 and 5 animals, respectively). (**b**) H&E staining of tumour masses (scale bars = 20 μm; 40× magnification, n = necrosis). (**c**) Representative images obtained after PCNA immunostaining of the same tumour masses (scale bars = 40 μm; 20× magnification). (**d**) Quantitative analysis of tumour weight, tumour volume, and PCNA positive cells. (n > 5; * *p* < 0.05).

**Figure 2 biomedicines-10-01784-f002:**
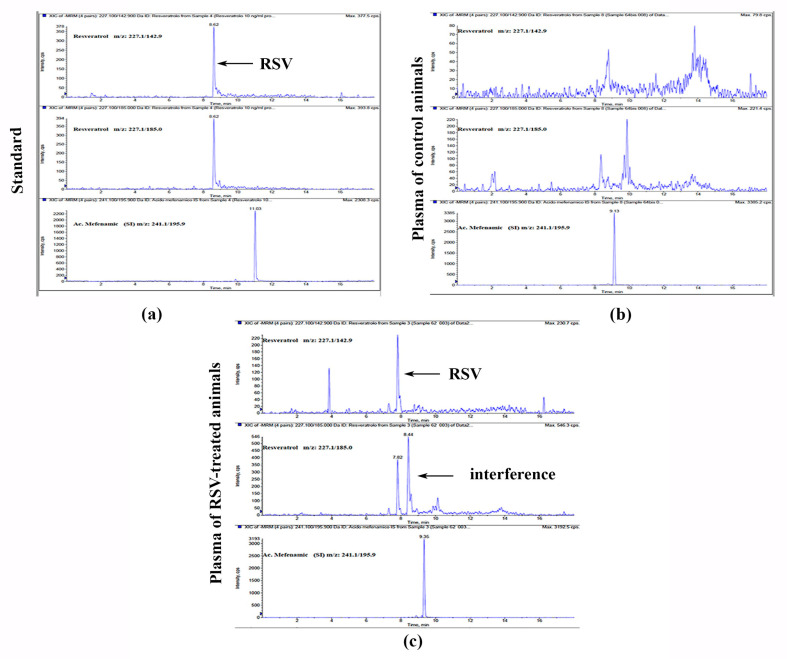
HPLC-MS/MS extracted chromatograms for RSV and Mefenamic acid (IS) of: (**a**) standard solution at 10 ng/mL concentration; (**b**) plasma of control mice; (**c**) plasma of mice treated with RSV.

**Figure 3 biomedicines-10-01784-f003:**
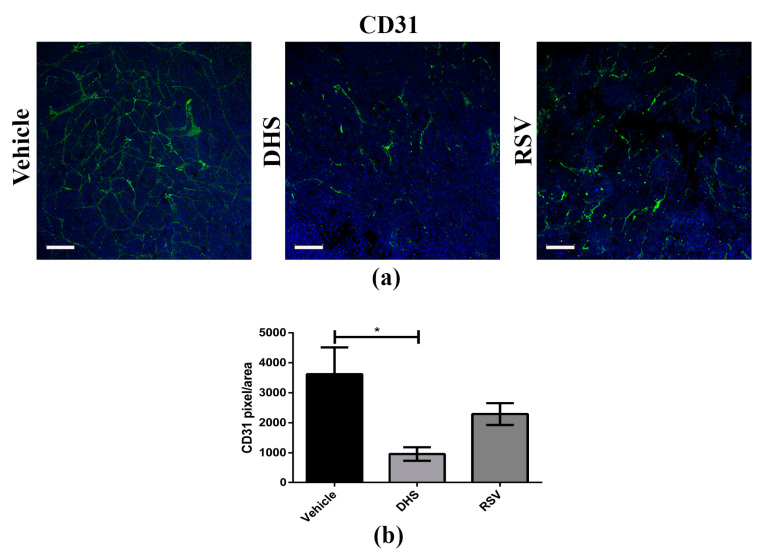
(**a**) Representative images of CD31 (green) whole mount staining (scale bar = 100 μm; 20× magnification); (**b**) Relative quantification of signal obtained by confocal microscopy (* *p* < 0.05).

**Figure 4 biomedicines-10-01784-f004:**
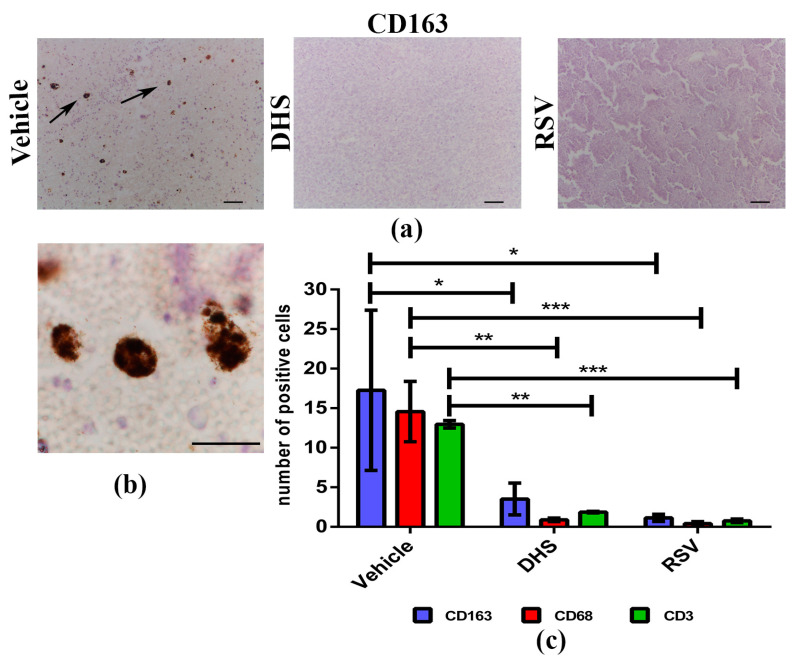
(**a**,**b**) Representative images of CD163 immunostaining in vehicle, DHS, and RSV tumour masses (scale bars = 100 μm; 20× magnification (**a**)) and in vehicle sample (scale bar 50 μm; 100× magnification (**b**)); (**c**) Quantitative analysis of CD163, CD68 and CD3 positive cells/sample (* *p* < 0.05; ** *p* < 0.01; *** *p* < 0.001).

**Figure 5 biomedicines-10-01784-f005:**
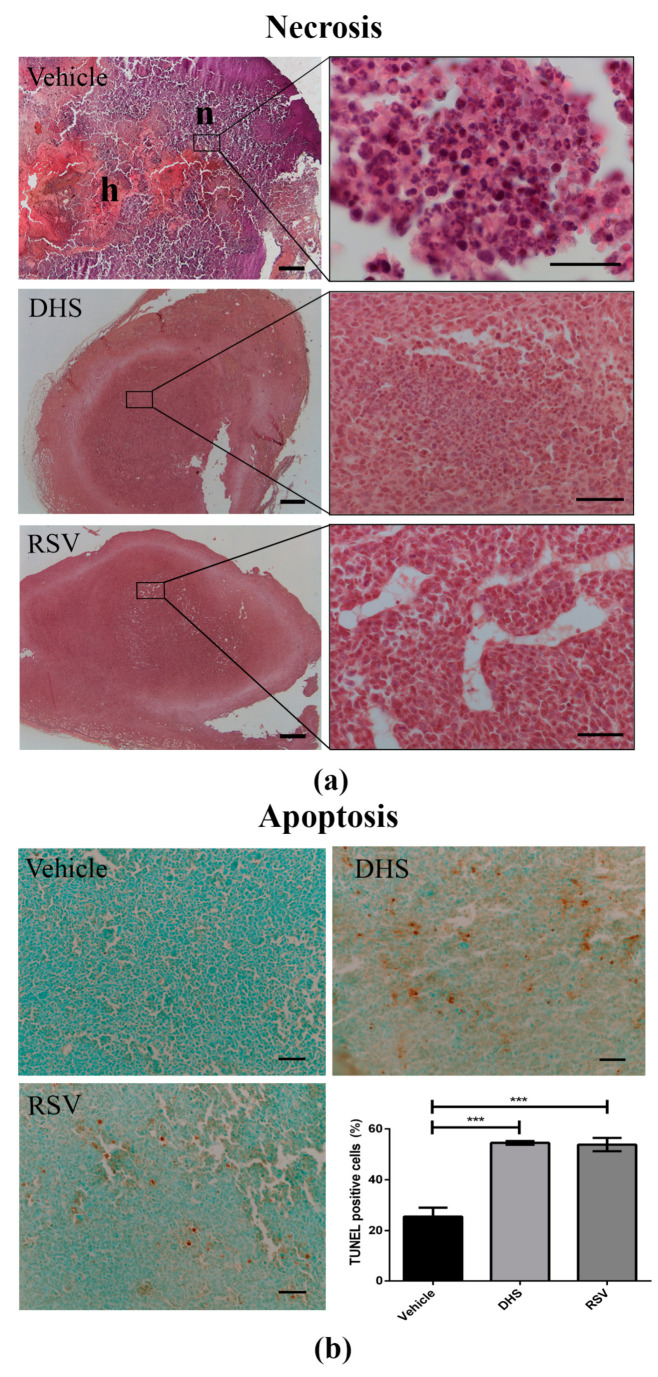
(**a**) Representative photomicrographs of H&E staining for vehicle, DHS, and RSV tumour masses (scale bars = 1000 μm; 4× magnification (left) and 100× magnification (right), n = necrosis, h = haemorrhage); (**b**) Representative photomicrographs of immunohistochemical staining for apoptotic cells in vehicle, DHS, and RSV tumour masses (scale bars = 100 µm; 100× magnification), and relative quantitative analysis (*** *p* < 0.001).

**Figure 6 biomedicines-10-01784-f006:**
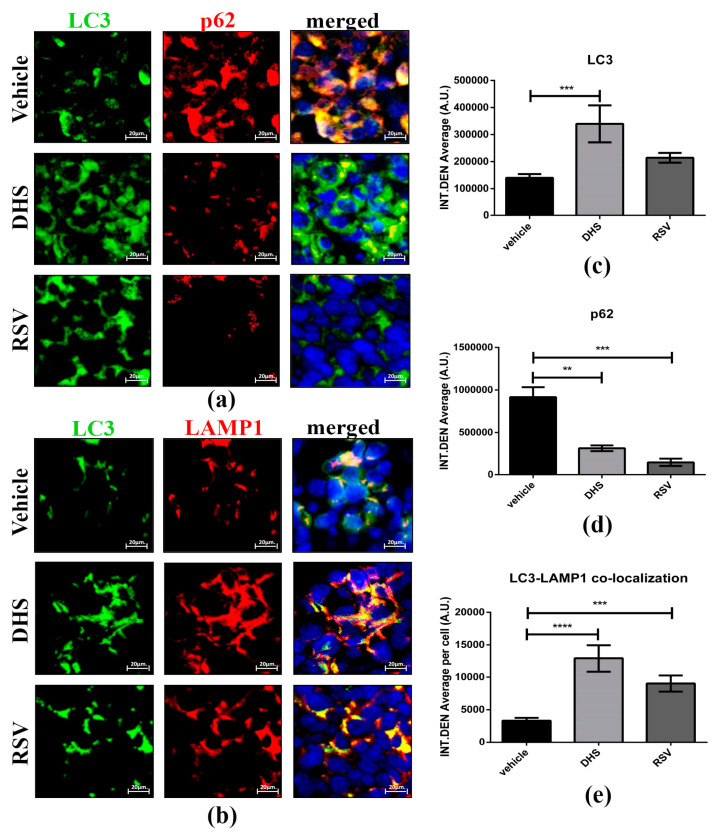
(**a**,**b)** Representative image of LC3 (green)/p62 (red) (**a**) and LC3 (green)/LAMP1 (red) (**b**) co-staining for vehicle, DHS-, and RSV-treated tumour masses (scale bars = 20 μm; 63× magnification) and relative quantification of LC3 (**c**), p62 (**d**) and LC3-LAMP1 co-localization signal (**e**) in cancer cells (** *p* < 0.01; *** *p* < 0.001; **** *p* < 0.0001).

**Figure 7 biomedicines-10-01784-f007:**
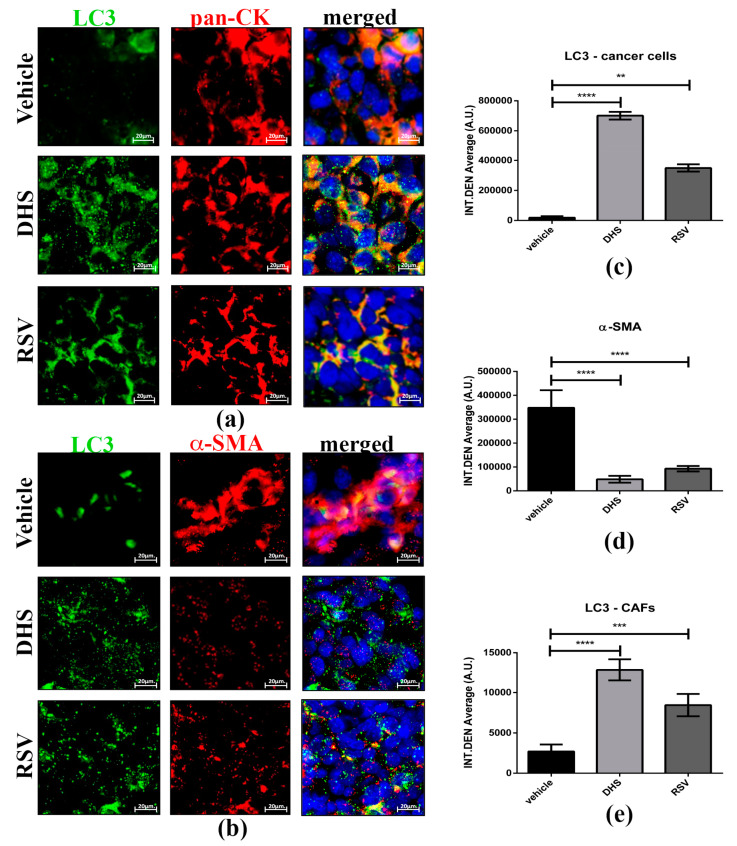
(**a**,**b**) Representative images of LC3 (green)/pan-cytokeratin (red) (**a**) and LC3 (green)/α-SMA (red); (**b**) co-staining for vehicle, DHS-, and RSV-treated tumour masses (scale bars = 20 μm; 63× magnification), and relative quantification of LC3 in pan-CK-positive epithelial cancer cells (**c**), while α-SMA (**d**) and LC3 levels were quantified in CAF population (**e**) fluorescent signals (** *p* < 0.01; *** *p* < 0.001; **** *p* < 0.0001).

## Data Availability

The data and underlying code presented in this study are available on request from the corresponding authors.

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
