# Peer review of "Resveratrol and Its Analogue 4,4′-Dihydroxy-trans-stilbene Inhibit Lewis Lung Carcinoma Growth In Vivo through Apoptosis, Autophagy and Modulation of the Tumour Microenvironment in a Murine Model"

_biomedicines, 2022, doi:10.3390/biomedicines10081784_

Round 1
Reviewer 1 Report
In the study titled “Resveratrol and its analogue 4,4'-dihydroxy-trans-stilbene in-2 hibit Lewis lung carcinoma growth in vivo through apoptosis, autophagy and modulation of the tumour microenvironment ” Ciro Isidoro, Lucia Anna Stivala and co-authors evaluate the anti-tumor effects of resveratrol (RSV) and its analogue 4,4’-dihydroxystilbene (DHS) in a lung cancer model represented by C57BL/6J mice bearing cancers resulting from Lung Lewis Cell (LLC) implantation. They assessed tumor mass weight, angiogenesis, cell proliferation and death, autophagy, immune microenvironment and infiltrating cancer-associated fibroblasts (CAFs) upon treatment for 21 days with RSV or DHS, solubilized in drinking water, one week before LLC implantation. They found that RSV and DHS hampered tumor growth by increasing apoptosis and autophagy and reduced tumor-associated immune-cells infiltration and CAFs maturation. The authors suggest that RSV or DHS could complement conventional chemotherapy
The paper contains new information.
The study design is clear and sound.
All potential biases have been appropriately addressed.
The methods are appropriate and well described, and sufficient details are provided.
There are no apparent errors of fact or logic.
The discussion and conclusions are well balanced and adequately supported by the data.
There are few typos. I recommend accurate proofreading of the manuscript
Reviewer 2 Report
The manuscript entitled “Resveratrol and its analogue 4,4'-dihydroxy-trans-stilbene inhibit Lewis lung carcinoma growth in vivo through apoptosis, autophagy and modulation of the tumour microenvironment” describes that both the stilbenes are able to contrast the growth of the LLC tumour masses by inhibiting the tumour-supporting features promoted by the components of the tumour microenvironment.
1. In the figure 1A, it shows no significantly difference among Vehicle, DHS and RSV groups.
2. In the figure 1C, it shows no significantly difference between Vehicle and RSV groups.
3. In mice model, authors should provide the numbers of treated-mice each group in the sections of Materials and Methods, and figure legend.
4. In the figure 4, authors should improve the quality the images. Authors should provide the images of CD68 and CD3 staining images.
5. In the figures 6 and 7, do authors measure these markers LC3, p62, LAMP1, pan-CK, alpha-SMA in which kind of cells, respectively?
6. It suggests to change “Resveratrol and its analogue 4,4'-dihydroxy-trans-stilbene inhibit Lewis lung carcinoma growth in vivo through apoptosis, autophagy and modulation of the tumour microenvironment” into “Resveratrol and its analogue 4,4'-dihydroxy-trans-stilbene inhibit Lewis lung carcinoma growth in vivo through apoptosis, autophagy and modulation of the tumour microenvironment in murine model”.
